# Solvent-non-solvent rapid-injection for preparing nanostructured materials from micelles to hydrogels

Chao Lang [1,2], Jacob A. LaNasa[1], Nyalaliska Utomo [1], Yifan Xu[1], Melissa J. Nelson[1], Woochul Song[2], Michael A. Hickner [1,3], Ralph H. Colby[1,3], Manish Kumar [2,3,4,5] & Robert J. Hickey [1,3]

Due to their distinctive molecular architecture, ABA triblock copolymers will undergo specific self-assembly processes into various nanostructures upon introduction into a B-block selective solvent. Although much of the focus in ABA triblock copolymer self-assembly has been on equilibrium nanostructures, little attention has been paid to the guiding principles of nanostructure formation during non-equilibrium processing conditions. Here we report a universal and quantitative method for fabricating and controlling ABA triblock copolymer hierarchical structures using solvent-non-solvent rapid-injection processing. Plasmonic nanocomposite hydrogels containing gold nanoparticles and hierarchically-ordered hydrogels exhibiting structural color can be assembled within one minute using this rapid-injection technique. Surprisingly, the rapid-injection hydrogels display superior mechanical properties compared with those of conventional ABA hydrogels. This work will allow for translation into technologically relevant areas such as drug delivery, tissue engineering, regenerative medicine, and soft robotics, in which structure and mechanical property precision are essential.

[1] Department of Materials Science & Engineering, The Pennsylvania State University, University Park, PA 16802, USA. [2] Department of Chemical Engineering, The Pennsylvania State University, University Park, PA 16802, USA. [3] Materials Research Institute, The Pennsylvania State University, University Park, PA 16802, USA. [4] Department of Biomedical Engineering, The Pennsylvania State University, University Park, PA 16802, USA. [5] Department of Civil and Environmental Engineering, The Pennsylvania State University, University Park, PA 16802, USA. Correspondence and requests for materials should be addressed to M.K. (email: manish.kumar@psu.edu) or to R.J.H. (email: rjh64@psu.edu)

Block polymers consist of two or more covalently attached incompatible polymer sequences, and as a result, will self-assemble into nanostructured materials with tunable morphologies and features on the 10–100 nm length scale[1]. The resulting nanoscale morphologies are a result of a balance between minimizing the different monomer contacts and maximizing the polymer block entropic degrees of freedom. Advances in synthetic strategies and chain architectural complexities have led to myriad applications for these materials ranging from uses in everyday products such as pressure-sensitive adhesives, coatings, nonwoven fabrics, and packaging[2–4], to materials for highly engineered products such as medical devices[5], organic electronics[6,7], separation membranes[8–10], and porous materials[11]. In addition to the progress in bulk nanostructural material fabrication using block polymers, similar studies of morphological control for solution-phase self-assembly have revolutionized areas of healthcare related to diagnostic detection and therapeutic administration[12–15]. Although the fundamental principles for equilibrium block polymer self-assembly in the melt and in solution are well-established, little attention has been paid on the guiding principles of nanostructure formation during nonequilibrium processing conditions, which arguably represents the most easily adaptable continuous manufacturing scenario for such materials.

In the immense block polymer architectural and compositional phase space[16], linear ABA triblock copolymers have been broadly used to form physically crosslinked gels for applications in a number of technologies like thermoplastic elastomers (TPE)[17,18], polymer electrolyte membranes[6,7], tissue engineering[19], and artificial muscles[20]. Gels, or hydrogels, are typically formed by means of polymer external stimuli[21], coacervation[22,23] or by simply submerging a neat ABA triblock copolymer into a B-selective solvent (e.g., water for hydrogels)[24,25]. The physical crosslinks in hydrogels created from ABA triblock copolymers form when the hydrophobic A-blocks segregate into separate micellar domains, while the mid B-block bridges the two domains (Fig. 1)[26]. As a result, the hydrophobic polymers will form interconnected domains supporting the hydrogel during swelling in aqueous environments. Additionally, because self-assembly is used to create the hydrogel, disassembly of the polymeric network

by use of external stimuli like temperature, salt concentration, and pH is possible, allowing for further processing or recycling[27]. At low ABA triblock copolymer concentrations in a B-selective solvent, discrete isolated spherical micelles (i.e., flower-like micelles) (Fig. 1) form where the two hydrophobic A-blocks will reside in the same spherical domain, and the mid B-block will create a loop[28]. The transition from flower-like micelles to physically crosslinked gels as concentration increases is defined as the critical gelation concentration (CGC)[25,26]. This transition is predicted to be governed by an increase in polymer bridging as the polymer aggregation number (number of polymer chains per micelle) increases with increasing concentration[26]. Bridging is favored over looping because overcrowding of polymer chains within a single micelle is relieved when a single chain spans two micelles[26].

Much of the existing literature regarding solution-phase self-assembly focuses on the thermodynamic equilibrium state of the system[29–32]. Despite the thermodynamic understanding of self-assembly, determining the guiding principles and establishing the essential parameters that dictate the final structure of ABA triblock copolymer solution-phase self-assembly under nonequilibrium conditions have yet to be fully addressed[33]. Traditionally, slow evaporation of the cosolvent[34] or solvent switching during water addition[35] to induce hydrogel formation, or swelling annealed structures in selective solvents[24,36] has been utilized to create nanostructured gels. Quickly forming structures in water using a rapid-injection method to induce self-assembly is desirable, more convenient, and commercially applicable in the development and implementation of nanostructured materials. While the benefits of forming nanostructured ABA-type hydrogels under commercially relevant processing conditions are preferable[37], the factors and experimental parameters that govern the gel formation under nonequilibrium conditions have yet to be fully investigated.

Here, we address the experimental parameters necessary for forming hydrogels using a rapid-injection method for a series of ABA triblock copolymers exhibiting hydrophobic−hydrophilic−hydrophobic chain sequences. We establish a quantitative route to obtain ABA triblock copolymer micelles, microgels, and hydrogels using a rapid-injection method. The final state of the

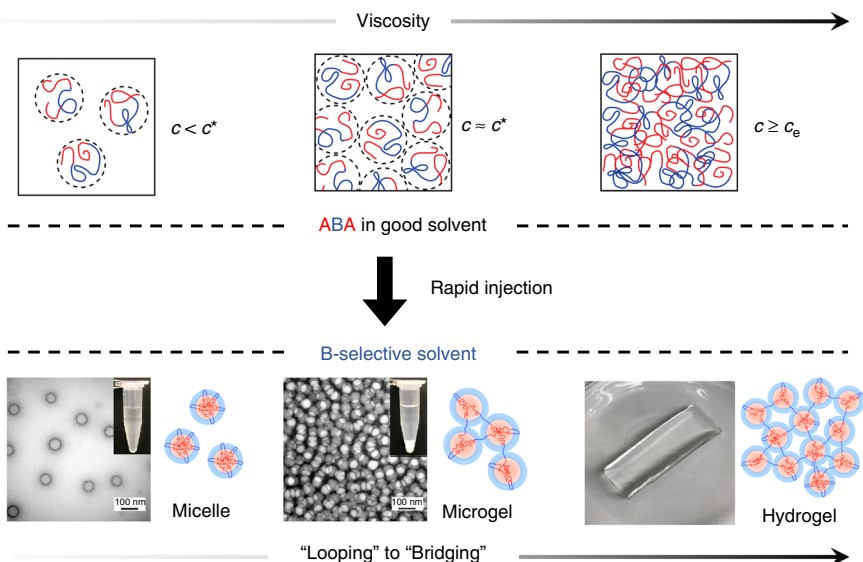

**Fig. 1** Self-assembly of ABA triblock copolymers via rapid-injection. The final state of the ABA triblock copolymer (micelles, microgels, and hydrogels) in a B-selective solvent is controlled and understood through the initial concentration of the polymer in the pre-injection solution. The self-assembly products ranging from micelles, microgels, and hydrogels correspond to dilute, semi-dilute, and entanglement regimes, respectively

ABA triblock copolymers (micelles, microgels, and hydrogels) depends on the initial polymer concentration, relative to the overlap concentration ($c^\star$), in a water miscible good solvent (e.g., tetrahydrofuran (THF)) before being rapidly injected into water (Fig. 1). At low concentrations in the dilute regime ($c < c^\star$), we produce micelles after injecting a polymer solution initially in THF into water. At concentrations near the overlap concentration ($c \approx c^\star$), we form microgels, which are discrete micrometer sized physically crosslinked micellar aggregates. For concentrations approaching the entanglement regime ($c \geq c_e$), the ABA polymer forms a hydrogel in which bridged hydrophobic domains span the entire sample volume. By rapidly injecting a polymer solution containing an amphiphilic triblock copolymer into an unfavorable B-selective solvent, the polymers rapidly self-assemble and form nanostructured products within 1 min. The ability to rationally design a variety of nanostructures with various ABA triblock copolymers by simply calculating $c^\star$ using the radius of gyration ($R_g$) of the polymer in the cosolvent emphasizes the versatility and universality of this self-assembly process. Interestingly, the amphiphilic triblock copolymer hydrogels created using the rapid-injection method exhibit hierarchical ordering in which spherical micelles form at the nanoscale, and a cellular network structure is composed of the physically crosslinked micelles at larger length scales. As a result of the well-defined hydrophobic domains (micelle cores) within the hydrophilic matrix, we are able to create printable structures, fibers, and coatings exhibiting fascinating properties that include the storage and retention of hydrophobic plasmonic gold nanoparticles and structural coloration. Furthermore, the hydrogel materials are easily recyclable by redissolution in the good solvent and exhibit the same mechanical and self-assembly properties as the original material after reprocessing. The work presented here will lead to broad implementation of ABA triblock copolymers in the creation of functional hydrogels with applications in drug delivery[38], tissue engineering[39], and personal care products[40].

## Results

**Relating initial polymer solution to the final state.** Four different amphiphilic triblock copolymers composed of poly (styrene)-poly(ethylene oxide)-poly(styrene) (SOS), poly(isoprene)-poly(ethylene oxide)-poly(isoprene) (IOI), and poly (butadiene)-poly(ethylene oxide)-poly(butadiene) (BOB) triblock copolymers) (Fig. 2a) with varying molecular weights and midblock O volume fractions ($f_O$) (Table 1) were used to demonstrate the versatility of the solvent-induced nanostructure formation of ABA triblock copolymer using the rapid-injection method. By using this series of ABA triblock copolymers, it was possible to determine the necessary experimental parameters that govern the self-assembly behavior of ABA triblock copolymers under nonequilibrium conditions. The triblock copolymers were synthesized in two general steps: (1) using a combination of sequential living anionic polymerization to afford the diblock copolymers and (2) coupling the diblock copolymers using 1,4-bis (bromomethyl)benzene to afford the resulting triblock copolymers (Supplementary Figs. 1–2, Supplementary Table 1). We chose the midblock O as the hydrophilic block due to potential biomedical applications[41], and uses in battery and transistor technologies[6,7]. For the hydrophobic blocks, low glass transition temperature ($T_g$) polymers I and B, and glassy S were selected to investigate the role $T_g$ played in the self-assembly and the mechanical properties.

Aqueous colloidal or hydrogel structures are formed by first dissolving the desired ABA triblock copolymer in THF (good solvent, Supplementary Table 2) at specific concentrations, and

then rapidly injecting the THF solution into water (1 mL of solution injected into 20 mL of water in 20 s). As seen in Fig. 2, the final structure of the SOS(8-65-8) triblock copolymer after rapidly injecting the polymer solution into water is easily tuned from micelles (Fig. 2b, Supplementary Table 3) to microgels (Fig. 2c) to hydrogels (Fig. 2d). When the polymer concentration in THF for SOS(8-65-8) was 0.02 wt%, the resulting structure after rapid-injection into water were isolated micelles (Fig. 2b). At a 1.0 wt% polymer solution concentration in THF, physically crosslinked micellar aggregates of colloidal size, or microgels formed after rapid-injection (Fig. 2c). To gain a better understanding of the transition from micelle to microgel, dynamic light scattering (DLS) was performed on samples using SOS(8-65-8) for a range concentrations near the overlap concentration. The DLS traces in Fig. 2e show that at the lowest concentration (0.05 wt% of polymer in THF) micelles form with an average hydrodynamic diameter of 100 nm. With increasing polymer concentration in the preinjection solution, two colloidal populations form: micelles and microgels. The DLS peak associated with the single micelles indicates that the micelle diameter stays relatively constant with increasing concentration. The single micelle size variations seen in DLS for 0.05 and 0.4 wt% samples are attributed to a fitting artifact of the intensity autocorrelation function when there is more than one population size, which is confirmed using TEM characterization (Supplementary Fig. 5). The average size of the microgel formed through physically crosslinked micelles increases from approximately 200 nm (0.4 wt%) to 1 μm (2.6 wt%). When the preinjection solution concentration reaches the entanglement regime, well-developed hydrogels form on rapid-injection.

The results shown in Fig. 2 are representative for the four different ABA triblock copolymers used for the rapid-injection method (Supplementary Figs. 3, 4). The major difference between the various polymer samples was the relationship between the initial polymer solution concentration in THF and the resulting polymer state in water. For example, the polymer sample with the lowest molecular weight, IOI(17-53-17), an initial polymer concentration of 1.5 wt% was needed to form microgels. Whereas for SOS(69-156-69), which is the polymer with the highest molecular weight, microgels formed when the initial polymer concentration in THF was 0.5 wt%. It is worth noting that when the IOI(17-53-17) polymer concentration was 0.5 wt%, only discrete micelles formed.

The concentration dependence of the final state of the self-assembled structures for a single ABA triblock copolymer, and the differences in the concentrations needed to form hydrogels for the various ABA triblock copolymers, suggests that the initial polymer solution properties in THF dictate the resulting structures after rapid-injection into water. To determine the role of the initial polymer solution properties in controlling the final structure, viscosity measurements were conducted at various concentrations for the four different polymers to experimentally determine the dilute, semi-dilute, and entangled regimes (Supplementary Fig. 6). When the specific viscosity ($\eta_{sp}$, where $\eta_{sp} = \eta_{solution}/\eta_{solvent} - 1$) is plotted versus polymer solution concentration ($c$), there are clear changes in the scaling exponents in the different polymer solution regimes[42,43]. In the dilute, semi-dilute, and entangled regimes, the scaling values are predicted to be 1, 1.3, and 3.9, respectively[42]. The fact that the experimentally determined scaling values for the four different ABA triblock copolymers follow the same trend as the theoretically predicted values indicate that there is no polymer association in the THF solution and the polymers are fully dissolved for all concentrations used[44]. When comparing the resulting structures with respect to initial concentration for SOS(8-65-8), we find that micelles, microgels, and hydrogels form when the polymer

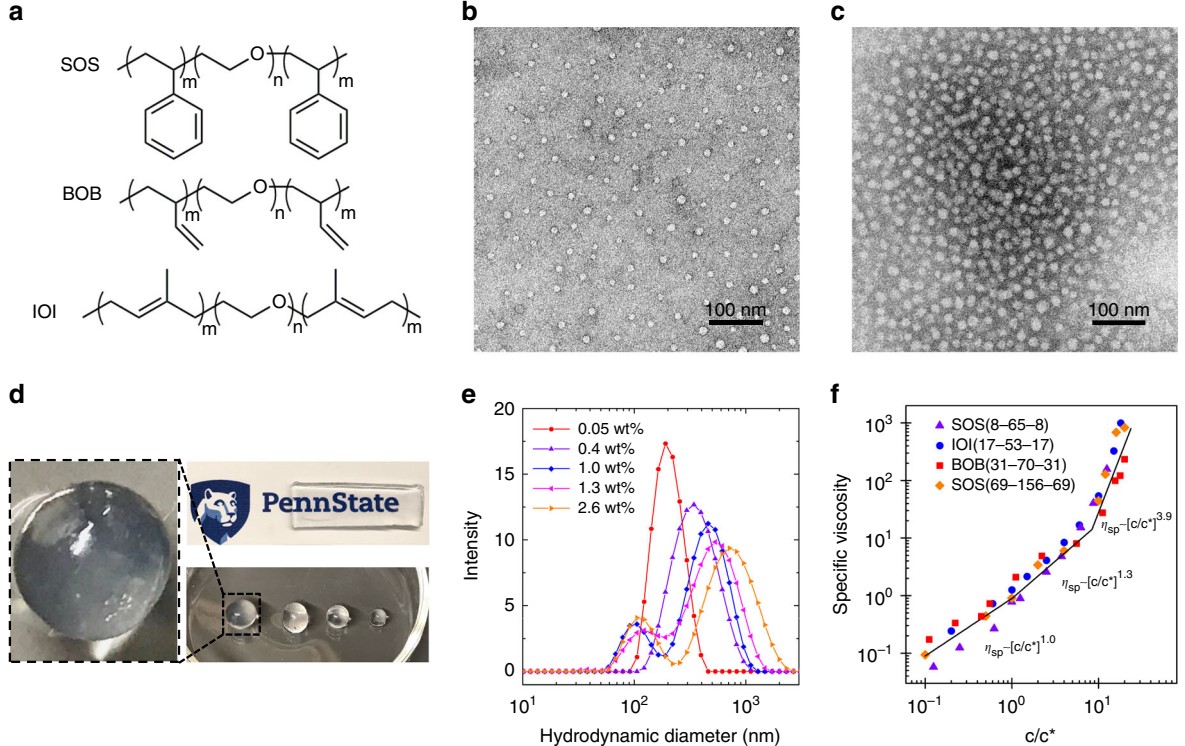

**Fig. 2** Polymer structure, morphology characterization, and viscosity measurements. **a** Chemical structures for the SOS, IOI, and BOB triblock copolymers used in this work. **b**, **c** TEM images of the distinctly different final aqueous states formed via rapid-injection using SOS(8-65-8). **b** Isolated micelles form when using a 0.02 wt% SOS(8-65-8) polymer solution in THF, and **c** microgels form when using a 1.0 wt% SOS(8-65-8) polymer solution in THF. The TEM images were negatively stained with 0.75% uranyl formate, which show the micelle cores in the dry state. The micelle size determined from TEM is less than the hydrodynamic radius observed by DLS. **d** Transparent spherical and rectangular hydrogels formed using a 10.0 wt% SOS(8-65-8) polymer solution in THF. The final hydrogel has a polymer content of 5.2 wt% in water (Supplementary Table 5). **e** DLS plots of the colloidal micelles and microgels formed when SOS(8-65-8) polymer solutions at different concentrations are injected into water. For microgel samples, DLS measurements were run on the supernatant after large aggregates were allowed to settle. **f** Universal specific viscosity versus normalized polymer concentration ($c/c^*$) plot for the SOS, IOI, and BOB polymers used in the study. The black lines indicate the expected power laws for dilute, semi-dilute, and entangled concentration regimes. Source data are provided as a Source Data file. TEM transmission electron microscopy, DLS dynamic light scattering, THF tetrahydrofuran

---

**Table 1 Triblock copolymer molecular weights, dispersity values, O volume fraction, and experimentally determined and calculated overlap concentration**

| Sample[a] | $M_{n,total}$[b] (kg mol$^{-1}$) | Đ[c] | $f_O$[d] | $R_g$[e] (nm) | $c^*_{exp}$ (wt%) | $c^*_{cal}$ (wt%) |
|---|---|---|---|---|---|---|
| SOS(8-65-8) | 68.8 | 1.03 | 0.80 | 14.5 | 0.8 | 1.0 |
| SOS(69-156-69) | 222.9 | 1.04 | 0.51 | 24.8 | 0.5 | 0.6 |
| IOI(17-53-17) | 54.5 | 1.05 | 0.55 | 11.9 | 1.0 | 1.4 |
| BOB(31-70-31) | 120.3 | 1.05 | 0.46 | 19.6 | 0.45 | 0.7 |

*NMR nuclear magnetic resonance, THF tetrahydrofuran*
[a]ABA(*m-n-m*) stands for triblock copolymer with A and B block number-average molecular weights of *m* kg mol$^{-1}$ and *n* kg mol$^{-1}$. S, B, I, and O are abbreviations for poly(styrene), poly(butadiene), poly (isoprene), and poly(ethylene oxide), respectively
[b]Total number-average molecular weight was determined from size-exclusion chromatography (SEC). Due to diblock copolymer residual, the measured $M_{n,total}$ of triblock copolymer is smaller than the theoretical values (twice of the precursor diblock copolymer $M_{n,total}$)
[c]Dispersity index ($M_w/M_n$) was determined using SEC
[d]O volume fraction was calculated using the $^1$H NMR data. Polymer density values were obtained from Sigma-Aldrich: I as 0.906 g mL$^{-1}$, S as 1.04 g mL$^{-1}$, B as 0.86 g mL$^{-1}$, and O as 1.13 g mL$^{-1}$ at 25 °C
[e]$R_g$ values were determined using a Wyatt Multi-Angle Light Scattering detector that was attached to the SEC, which was run with THF as the mobile phase

---

solutions are in the dilute, semi-dilute, and entangled regimes, respectively. When all of the $\eta_{sp}$ values for the four polymers are compiled into a single master plot, where the polymer concentration is normalized to the experimentally determined overlap concentration ($c/c^*$, where $c^*$ is $\eta_{sp} = 1$), the same three polymer solution regimes are established (Fig. 2f), and all four polymers are consistent with the trend that micelles, microgels, and hydrogels form when the initial polymer solution concentration in THF increases from dilute, to semi-dilute, and finally entangled regimes, respectively. It is worth noting that the

transition between dilute and the semi-dilute regimes results in a mixture of discrete isolated micelles and microgels (Fig. 2e).

By establishing the relationship between the initial concentration of the polymer solution and the final state of the assembled polymer structure in water, it is now possible to design desired structures for any ABA triblock copolymer, where the A-blocks are hydrophobic, and the molecular weight is sufficiently large to exhibit the three polymer solution regimes. Furthermore, the predictive nature of using the polymer solution properties allows one to easily calculate $c^*$ by using the $R_g$ of the ABA triblock

copolymer in THF (Supplementary Note 1). The overlap concentration is a way to determine the concentration ranges for the different regimes (dilute regime: $c < c^*$, semi-dilute regime: $c^* \leq c < c_e$, and the entangled regime: $c \geq c_e$), and can be calculated using Eq. 1.

$$c^* = \frac{M_n}{VN_A}.  \quad (1)$$

In Eq. 1, $M_n$, $V$, and $N_A$ are the polymer number-average molecular weight, the pervaded volume, and Avogadro's number. The utility of calculating $c^*$ using Eq. 1 is exemplified for the SOS (8-65-8) sample. When we use the experimentally measured $R_g$ (Table 1) to calculate $V$ ($V = \frac{4}{3}\pi R_g^3$), we obtain $c^* = 1.0$ wt% (Supplementary Table 4), which is close to the $c^*$ value determined from $\eta_{sp}$ (Fig. 2f). In the case of the hydrogel formation, polymer solutions need to be in the $c \geq c_e$ regime, which is approximately ten times $c^*$ [43,45]. For the four polymers studied here, the normalized $\eta_{sp}$ plot (Fig. 2f) indicated that the crossover from the semi-dilute to the entangled regime occurs at ~$10c^*$.

**Structure and mechanical properties of the hydrogels.** Remarkably, for the ABA triblock copolymer hydrogels fabricated using the rapid-injection process described above, the internal structure of the material exhibits hierarchical ordering in which the hydrophobic domains self-assemble at the nanoscale while the polymer forms a cellular network structure. A combination of cryogenic scanning electron microscopy (cryo-SEM) and small-angle X-ray scattering (SAXS) were used to investigate the internal structure of the hydrogels. Cryo-SEM images reveal that all the hydrogel samples (SOS, BOB, IOI) formed using rapid-injection contain spherical micelles (Fig. 3a, Supplementary Fig. 7). As expected, the diameter of the micelle core changes as the polymer molecular weight and composition changes (80 nm for SOS(69-156-69) compared with 58 nm for SOS(8-65-8), Fig. 3a, Supplementary Fig. 7a). The micelle core sizes determined from cryo-SEM are larger than the micelle cores measured from TEM, which is likely due to the sample characterization in the dry (TEM) and swollen (cryo-SEM) states, and the difficulty in distinguishing isolated micelles from microgel aggregates using cryo-SEM. Remarkably, the hydrogels exhibit a hierarchical structure in which nanostructured polymer layers form a cellular structure that surrounds water cavities. The cellular structure was discovered by comparing cryo-SEM images of the same SOS(69-156-69) hydrogel sample before and after vitrification (Supplementary Fig. 7a and Fig. 3b). The nanostructure for both IOI(17-53-17) and BOB(31-70-31) samples was confirmed using SAXS (Fig. 3c). SAXS patterns for the SOS samples were not analyzed due to low scattering contrast between the poly(styrene) micelle core and water/poly(ethylene oxide) midblock (Supplementary Table 6), which is exacerbated using the lab-source SAXS instrument. The SAXS patterns for both IOI(17-53-17) and BOB(31-70-31) were further interpreted by simulating the scattering pattern with a combination of a spherical form factor and a Percus–Yevick structure factor (Supplementary Note 2, Supplementary Eqs. 4−9) [24,46]. In both cases, the center-to-center distance of the micelles were assumed to be larger than the nanosphere radius of the micelle core (Supplementary Table 7). With this assumption, modeled SAXS profiles estimate that for the IOI(17-53-17) sample contains micelles with a micelle core radius of 12 nm and an inter-micellar center-to-center spacing of 47.6 nm. The hydrogels formed using BOB(31-70-31) exhibit a micelle core radius and an inter-micellar center-to-center spacing of 27.5 and 86 nm, respectively. The increase in micelle diameter and spacing for BOB over IOI is consistent with the reported molecular weight

of both samples and TEM results (Table 1, Supplementary Table 3). Furthermore, the simulated SAXS patterns suggest that the micellar core volume fraction of the IOI hydrogel and BOB hydrogel samples are greater than 35%, which is significantly larger than the experimentally determined polymer volume fraction (5.9% for IOI and 7.0% for BOB determined using mass differences in the swollen and dried states). We posit that the discrepancy in micellar volume fraction is indicative of the hierarchical structure of the hydrogel seen in Fig. 3b. The $q$-range in which the SAXS characterization was performed would be probing the wall structure (nanometer), not the larger length scales associated with the cellular network (micrometer). It is worth noting that the SAXS patterns for microgels and hydrogels are similar (Supplementary Fig. 8), indicating that the nanostructures (micelle core diameter, inter-micellar spacing, and micelle core volume fractions) do not change as the network spans larger length scales. Morphologically, the microgels formed in the work presented here resemble the hydrogels at the nanometer scale. The difference between the two structures is the ratio between looping and bridging chains. The micron-sized micellar aggregates, or microgels, will have an increased number of looping chains, as compared to hydrogels formed at higher concentrations. As the concentration is increased, more bridging chains between different micelles form, which leads to more robust gels with higher water contents (Supplementary Table 5). Here, the microgel term is used to classify the transition between isolated micelles and fully developed hydrogels.

To quantify the mechanical properties of the physically crosslinked hydrogels, tensile tests were performed using hydrogel fiber samples ($0.45 \pm 0.13$ mm in diameter) prepared via rapid-injection (Supplementary Movie 1). ABA triblock copolymer hydrogels exhibited a range of tensile strengths, elongation at breaks, and moduli (Supplementary Fig. 9), presenting the possibility to tune the mechanical properties of the material through careful selection of polymer species, molecular weight, volume fraction, and hydrophobic physical properties (e.g., $T_g$) (Fig. 3d–f, Supplementary Table 8). Comparison of BOB and SOS hydrogels show that using glassy hydrophobic micelle cores as physical crosslinks (SOS samples) enhances both the fracture energy and elongation capacity of the samples (Fig. 3d). With rubbery blocks forming the hydrophobic domains, BOB hydrogels showed limited strength (fracture energy of ~300 J m$^{-2}$) and low elongation at break (~1.1). Using the same polymer concentration (10 wt%) of SOS(69-156-69) as compared to BOB hydrogels, the prepared hydrogel showed significantly increased fracture energy of ~2000 J m$^{-2}$ and elongation at break of ~5. By further increasing the O volume fraction from 51 to 80%, hydrogels made from SOS(8-65-8) showed strain-stiffening properties [47–50]. Strain-stiffening has been observed in many physically associated polymer network systems [47–50]. Two main theories regarding the strain-stiffening mechanism have been proposed: (1) failure of the Gaussian polymer chain assumption for the midblock at large strain values [47,48,51] and (2) increased number of intermolecular associations leading to an enhancement in the number of elastically effective chains between physical crosslinks during stretching [48,52]. A more systematic study is needed to determine the exact mechanism governing the nonlinear mechanical behaviors of the gel network created by rapid-injection, and the role the hierarchical morphology plays. By increasing the initial polymer solution concentration from 10 to 15 wt% for hydrogels composed of SOS(8-65-8), there was an increase in the strain at break (~7.5) and the fracture energy (~3000 J m$^{-2}$). The mechanical properties of the SOS(8-65-8) hydrogels were also found to be strain rate dependent, as elongation (Supplementary Fig. 10) and fracture energy were drastically increased as strain

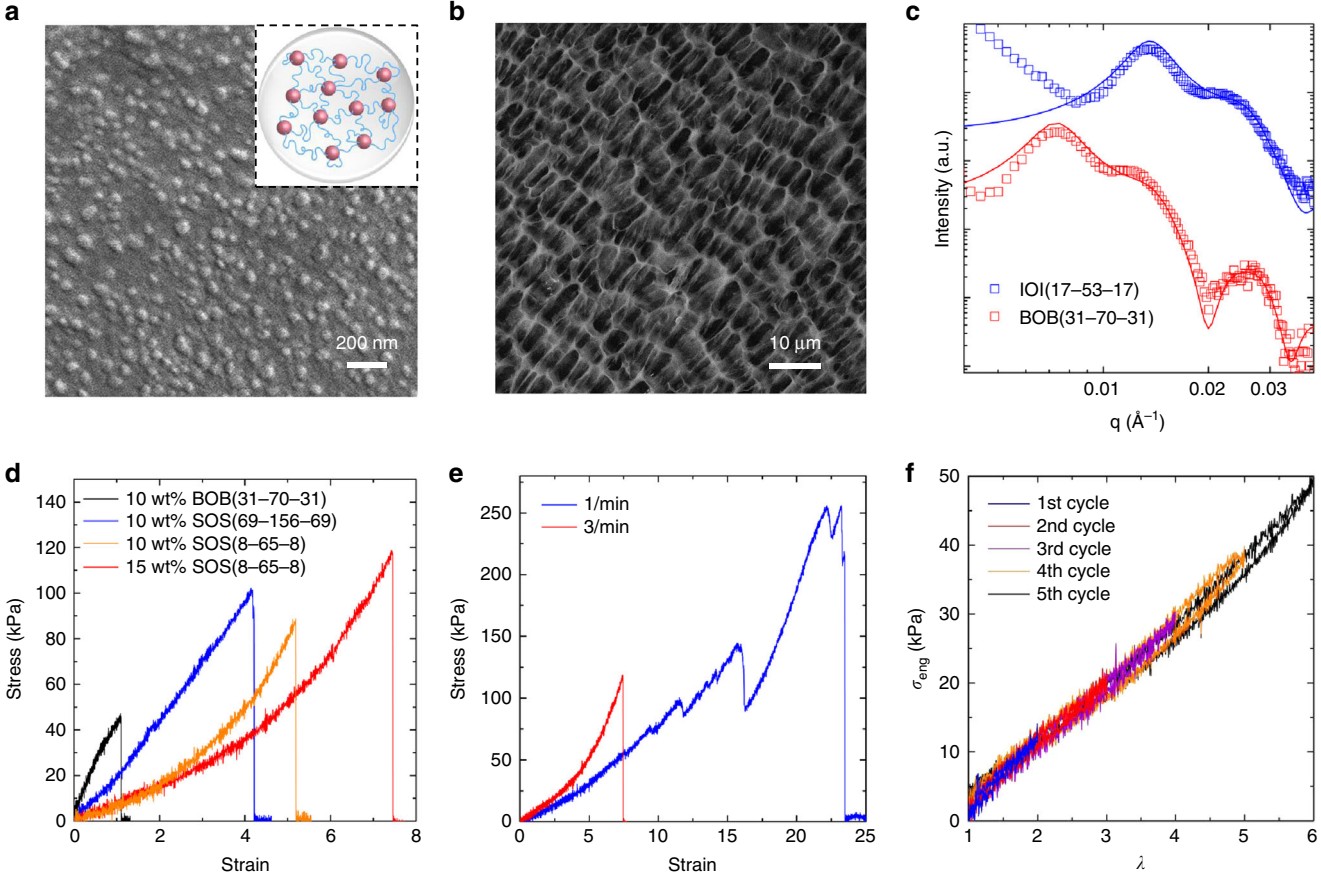

**Fig. 3** Nanostructural and mechanical property characterization of hydrogels. **a** Cryo-SEM image showing the self-assembled spherical micellar structure of the hydrogel formed using rapid-injection of a 10 wt% SOS(8-65-8) THF solution. **b** Cryo-SEM image of hydrogel sample from rapid-injection of a 10 wt% SOS(69-156-69) THF solution after sublimation showing the cellular network structure. **c** Experimental and simulated SAXS profiles for hydrogel samples using 10 wt% BOB(31-70-31) and 15 wt% IOI(17-53-17) THF solutions. **d** Stress-strain curves of hydrogels from using different triblock copolymer THF solutions. The strain rate used was 3 min$^{-1}$. Mechanical properties obtained from the curves are compiled in Supplementary Table 8. **e** Stress−strain curves of hydrogel from using a 15 wt% SOS(69-156-69) THF solutions measured at different strain rates. Under lower strain rates, the hydrogel can be stretched beyond 23 times of its original length. At large strain values, the stress−strain curve shows relaxation steps, suggesting possible occurrence of internal structure rearrangement and reformation. **f** Cyclic loading and unloading curves of hydrogel from using a 15 wt% SOS(8-65-8) THF solution at different elongation values. The strain rate used was 3 min$^{-1}$. Source data are provided as a Source Data file. THF tetrahydrofuran, SEM scanning electron microscopy, SAXS small-angle X-ray scattering

rate was reduced from 3 to 1 min$^{-1}$ (Fig. 3e). At low strain rate, the polymer chains in the hydrogel networks have more time to adapt to deformation, and as a result, exhibit increased elongation at break values. Unexpectedly, the stress−strain curve shows multiple relaxation steps at low strain rates (Fig. 3e). Although replicate experiments show these relaxation steps are reproducible and occur at similar strain values (Supplementary Fig. 11), the exact cause of these steps is not currently known and further in-depth studies are needed to fully address the mechanism[53,54]. The resilience of the hydrogel was further exemplified with cyclic loading−unloading tests (Fig. 3f). When a strain rate of 3 min$^{-1}$ was applied, the hydrogel is fully recoverable up to six times of its original length. At larger stain values ($\varepsilon > 3$) hysteresis is observed in the stress−strain curves. The ABA triblock copolymer hydrogels formed using the rapid-injection method display exceptional mechanical properties that are a result of the weak hydrophobic interactions at the nanoscale and hierarchically structured cellular network in the microscale. Hydrogel materials exhibiting superior toughness and stretchability typically require elaborate design and multiple components[55], such as use of engineered proteins and polypeptides[56], multiple crosslinking networks[57], and host−guest interaction-based molecular

sliding[58]. The rapid-injection method reported here represents a simple and versatile method for preparing hydrogel materials with superior mechanical properties. An additional benefit using the rapid-injection method is that the hydrogels are fully recyclable after formation. By simply dissolving the hydrogels in THF, precipitating the material in an isopropanol/hexane mixture, and isolating the polymer, all ABA triblock copolymers were recovered in high quality without sacrificing the mechanical properties (Supplementary Fig. 2e, 2f), making this material ideal for reprocessing and recycling.

**Versatile processing methods for functional hydrogels.** The rapid-injection method and the design rules for forming hydrogels described in this work are easily translatable to a variety of processing procedures that include printing, forming fibers, and coating complex architectures. For example, the well-defined internal nanostructure and the hierarchically ordered cellular network of the hydrogels allows one to create plasmonic nanocomposite hydrogels containing gold nanoparticles and hydrogels exhibiting structural color (Fig. 4). The procedure for printing hydrogels includes first printing a polymer/THF solution on a

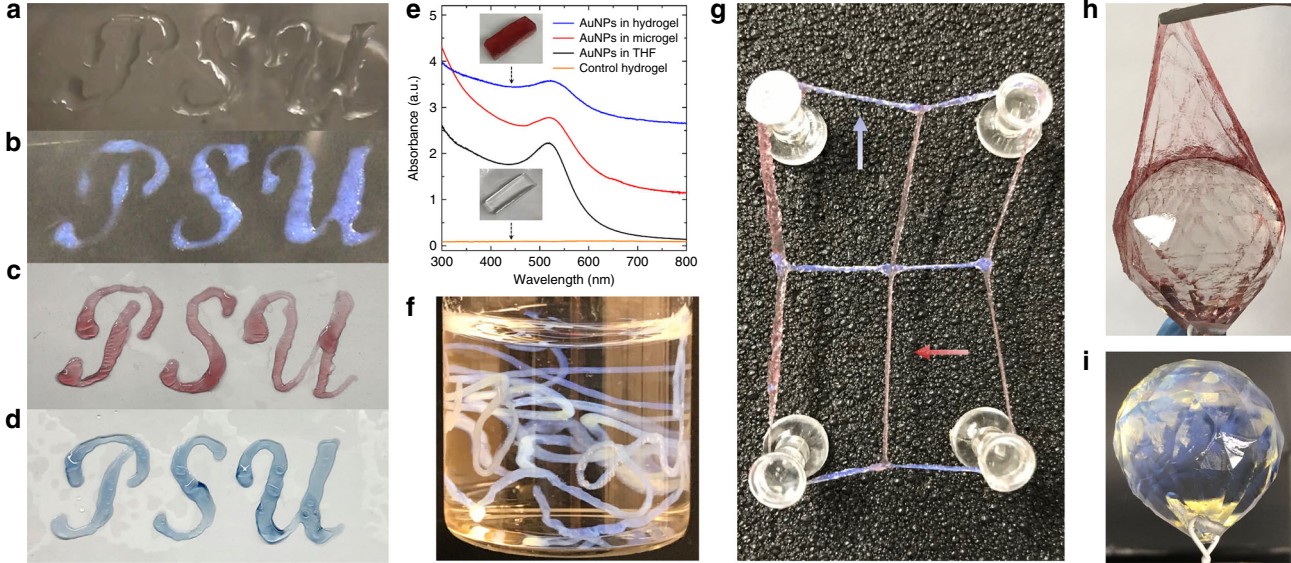

**Fig. 4** Hydrogels used for printing, fiber formation, and coatings. **a**–**d** Demonstration of hydrogel printing by submerging a pre-printed polymer/THF solution into water. **a** Colorless SOS(69-156-69)/THF solution before water submersion. **b** Development of structural color after immersing the printed SOS(69-156-69)/THF solution into water. **c**, **d** Printed hydrogels of SOS(8-65-8) hydrogel loaded with **c** gold nanoparticles (10 wt% gold with respect to the polymer) and **d** the hydrophobic dye indigo. **e** UV−Vis spectrum of AuNPs in THF, SOS(8-65-8) microgel loaded with AuNPs, SOS(8-65-8) hydrogel, and SOS(8-65-8) hydrogel loaded with AuNPs. The AuNPs-loaded hydrogel was prepared from rapid-injection of a THF solution of 10 wt% SOS(8-65-8) with AuNPs (10.0 wt% with respect to polymer). **f** Structural colored hydrogel fibers created by injecting a SOS(69-156-69)/THF solution into water. **g** Complex hydrogel mesh created by submerging a preprinted mesh using different polymer inks (SOS(69-156-69)/THF and SOS(8-65-8)/gold nanoparticle/THF solution) into water. In the mesh, structural colored hydrogel fibers are horizontally arranged (blue arrow) and plasmonic hydrogel fibers (red arrow) are vertically arranged. **h**, **i** Hydrogel coatings on a faceted glass sphere. The glass sphere is able to be coated with **h** a plasmonic hydrogel layer, or **i** a structural colored hydrogel layer. The coatings were fabricated by submerging a cleaned faceted glass sphere into **h** a THF solution of 10 wt% SOS(8-65-8) with AuNPs (10 wt% with respect to polymer), and **i** a THF solution 10 wt% SOS(69-156-69). Source data are provided as a Source Data file. THF tetrahydrofuran

substrate (glass slide) and then submerging the glass slide into water. Interestingly, printed hydrogels created using transparent SOS(69-156-69) polymer solution (Fig. 4a) develop structural color after 90 s once submerged in water (Fig. 4b, Supplementary Fig. 13, Supplementary Movie 2). The structural coloration of the hydrogel is a result of the material to reflect or scatter specific wavelengths of light depending on the morphology and the differences in the refractive indices of the different domains[59–61]. The structural color was not expected because the same polymer does not exhibit structural color in the microgel form (Supplementary Fig. 3b, inset), which is expected to have similar micelle diameter, inter-micellar spacing, and micelle volume fractions as in the hydrogel (Supplementary Fig. 8). We hypothesize that the SOS(69-156-69) polymer exhibits structural color due to the hierarchical cellular network structure although the nanostructural ordering may play a role as well. The structural coloration, and as a result, the self-assembly and gelation process, are easily monitored after adding the glass slide to water (Supplementary Movie 2). Additional hydrogel functionality is possible by adding hydrophobic dyes or hydrophobic gold nanoparticles into the initial polymer/THF solution. The internal nanostructure of the hydrogels allows one to easily add hydrophobic components (oleylamine-capped gold nanoparticles or indigo blue) into the hydrophobic micellar domains (Fig. 4c, d). The hydrophobic domains serve as stable residing sites that prevent nanoparticle aggregation; gold nanoparticles inside the hydrogels were shown to display the same plasmonic UV−Vis spectrum as gold nanoparticles dispersed in THF (Fig. 4e, Supplementary Fig. 12). The plasmonic properties of the gold nanoparticles give rise to the red color seen in the hydrogel materials, which is a result of the collective oscillation of electrons in the nanoparticle[62]. By co-

assembling gold nanoparticles and polymers using solution processing techniques, it is possible to create materials exhibiting physical properties that are dictated by the nanoparticle, which is one of the attractive features of colloidal nanoparticle/polymer materials[63–65].

Additional hydrogel structures including fibers and coatings illustrate the versatility of the processing method (Fig. 4f–i). A complex hydrogel mesh was made by printing with two different types of polymer solution inks (SOS(69-156-69)/THF and SOS(8-65-8)/gold nanoparticle/THF solution) onto a stainless steel substrate and then immersing the substrate into water. The exceptional strength of the mesh junction points formed by different polymer solutions were further highlighted by the overall elasticity of the mesh that was removed from the substrate (Fig. 4g, Supplementary Movie 3). Furthermore, structural colored hydrogel fibers are easily formed by directly adding the polymer/THF solution into water (Fig. 4f, Supplementary Movie 1). The advantage of forming hydrogel materials using a direct injection into water is the possibility for scaling up material fabrication, which is analogous to solvent- or gel-spinning polymer fibers in which highly aligned polymer fibers are created by adding a homopolymer solution to a coagulating liquid[66,67].

Another exciting application of hydrogels is film coating for applications in biocompatible interfaces for implants and devices in biological systems[68]. Similar to the printing process, hydrogel coating on an assortment of three-dimensional objects are created by dipping the objects into the polymer/THF solution and then submerging them into water (Fig. 4h, i, Supplementary Movie 4). Figure 4h, i shows how a faceted glass sphere is able to be coated with either a plasmonic or a structural colored hydrogel. In addition, it is possible to remove the coating in a single piece

(Fig. 4h). The fundamental understanding of the solvent-induced self-assembly process and the ability to apply the method to create a variety of hydrogel structures has limitless possibilities for creating ABA triblock copolymer nanostructured materials. An exciting prospective is that traditional polymer processing methods like solvent- or gel-spinning are now viable possibilities for expanding the applicability of block polymers in large-scale processing applications.

## Discussion

The solvent-induced nanostructural transition work presented here takes advantage of creating hydrogel materials using a highly nonequilibrium process of rapidly injecting an ABA polymer solution into a B-selective solvent. The rapid-injection method is a robust technique for preparing ABA triblock copolymer self-assembled materials with outstanding mechanical properties. The rapid-injection process allows one to easily tune structures (micelles, microgels, and hydrogels) by simply changing the initial polymer concentration. Specifically, the final state of the system is related to the initial polymer concentration in THF relative to $c^*$, which will allow one to utilize knowledge of $c^*$ to rationally design and tailor desired nanostructured hydrogel materials. Furthermore, the rapid-injection method is translatable for printing desired structures, forming fibers, and coating complex three-dimensional objects. The internal nanostructure and the hierarchically ordered cellular network structure of the hydrogels allows one to create plasmonic nanocomposite hydrogels containing gold nanoparticles and hydrogels exhibiting structural colors.

## Methods

**Synthesis and characterization of block copolymers**. Monomers (Sigma-Aldrich) were purified with either *n*-butyllithium (Sigma-Aldrich) (isoprene, 1,3-butadiene, and ethylene oxide) or di-*n*-butylmagnesium (Sigma-Aldrich) (styrene). Organic solvents were used directly from a solvent drying system (JC Meyer Solvent Systems). The monomers isoprene, 1,3-butadiene, and styrene were initiated using *sec*-butyllithium (Sigma-Aldrich). Potassium naphthalenide solutions used for the poly(ethylene oxide) polymerizations were prepared 24 h before the initiation of the polymerization. The nuclear magnetic resonance (NMR) experiments were performed on an Avance AV3HD 500 NMR spectrometer (Bruker) at room temperature. All the obtained spectra were calibrated according to the residual solvent peak. The size-exclusion chromatography (SEC) experiments were performed on an EcoSEC HLC-8320GPC (Tosoh Bioscience) equipped with a DAWN multi-angle static light scattering (SLS) detector (Wyatt Technology).

Mono-hydroxyl functionalized PEO, diblock copolymers SO, BO and IO were synthesized via standard anionic polymerization[69]. Triblock copolymers SOS, BOB and IOI were synthesized by terminating the corresponding diblock copolymers with the coupling agent α,α′-dibromo-*p*-xylene[70]. All the polymers were characterized with SEC and $^1$H NMR (see detailed data in supporting information).

**Micelle and microgel preparation and characterization**. Block polymers were first dissolved in THF in the dilute or semi-dilute concentration regimes. To perform self-assembly, 1 mL of the polymer solution was rapidly injected to 20 mL of water in 20 s at room temperature. Transmission electron microscopy (TEM) data were collected on an FEI Tecnai G2 Spirit BioTwin instrument operated at 120 kV. All the samples were prepared and negative stained with 0.75% aqueous uranyl formate solution. Dynamic light scattering (DLS) was performed on a Malvern instrument equipped with a HE-Ne laser (633 nm, 4 mW) and an avalanche photodiode detector. For microgel samples, the supernatant of settled solution was used for the measurement. The samples were equilibrated at 25 °C before the measurement and signals at 90° angel were collected.

**Viscosity measurements**. The viscosity values of different polymer solutions were measured on a Rheometric Scientific ARES RFS III strain-controlled rotational rheometer using concentric cylinder geometry with bob diameter of 16.5 mm. The temperature was controlled by a Julabo FS 18 recirculating water bath and evaporation was prevented by covering sample with a solvent trap with embedded felt. Polymer solutions (1.5 mL) were placed in the cylinder and a geometry gap of 3.5 mm was used to perform steady shear measurements. Steady shear measurements of solutions were conducted with shear rates of 10–100 s$^{-1}$, with the motor rotating in both directions. Steady-state sensing with an equilibration time of 10 s and an averaging time of 30 s was applied to the procedure to avoid measuring viscosity overshoot during start-up shear.

**Hydrogel preparation**. To prepare the hydrogels, ABA triblock copolymers were first dissolved in THF in the entangled regime. The solution was then injected into an excess amount of water (20 mL). For making bulk hydrogel samples and hydrogel printing, the polymer solution was first written on a substrate or cast in a petri dish, then an excess amount of water was added to the solution. For fabricating free-standing hydrogel fibers, the polymer solution was injected to water at a constant rate (5 mL min$^{-1}$) with a nozzle with an inner diameter of 0.51 mm. For coating applications, the substrate was first immersed into polymer solution and then submerged into water for several minutes until the hydrogel formed.

**Cryogenic scanning electron microscopy (cryo-SEM)**. The cryo-SEM experiments were performed on a Zeiss Sigma VP-FESEM. To perform the experiments, the hydrogel sample was placed together with cryo-matrix on the holder of the sample exchange rod. The sample holder was then submerged into a high-pressure freezing machine (slushy) filled with liquid nitrogen. After the sample was frozen, it was inserted into the SEM preparation chamber and a scalpel was used to cleave the sample and expose the inside of the hydrogel. Then the sample was sputter coated with gold for 120 s inside the chamber. Finally, the sample was inserted into the main chamber of the cryo-SEM maintained at −195 °C. The samples were measured under variable pressure mode at a low acceleration voltage of 5 keV to avoid excessive charging and radiation damage to the sample.

**Small-angle X-ray scattering (SAXS)**. Transmission SAXS measurements were performed using a Cu K-α sourced (1.54 Å and 8.04 keV) Xeuss 2.0 (XENOCS) instrument installed with collimation optics and a 2D X-ray detector Pilatus 200 K (Dectris). The sample to detector distance was set to 2.5 and isotropic scattering intensity were azimuthally integrated to obtain $I(q)$ 1D plots. The scattering wavevector, $q$, was calibrated by using a standard sample of powdered silver behenate. Microgel samples were measured in solution in quartz capillaries (1.5 mm outer diameter thickness, Charles Supper Company, Quartz 15-QZ). Hydrogel samples were measured after immediate removal from water to ensure a completely swollen network using an acquisition time of 5 min.

**Tensile test of ABA triblock copolymer hydrogels**. Uniaxial tensile stress−strain tests were performed on an Instron 5866 load frame with a 10 N load cell at 25 °C. To measure the mechanical properties of ABA triblock copolymer hydrogels, fibers with an average diameter of 0.45 ± 0.13 mm were prepared via rapid-injection. The initial gauge length was 10 mm and uniaxial extension was applied to the specimen at a constant rate. The samples were stretched until fracture, and measurements were conducted in triplicate for each sample. The stress−strain curves of the hydrogel samples were background subtracted. For the cyclic loading and unloading experiment, stress−strain curves were recorded for different strain values ranging from 1 to 5 for the same sample.

## Data availability

The relevant data are available from the authors upon reasonable request. The raw data for Figs. 2e, 2f, 3c, 3d, 3e, 3f and 4e and Supplementary Figs. 2, 4, 6, 8, 10, 13 are listed in a Source Data file.

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

## Acknowledgements

This work was supported using start-up funds and the Gladys Snyder Junior Faculty Grant from the Penn State University. Part of the work was also funded by National Science Foundation (NSF) grant CBET 1552571 and US Army CERL W9132T-16-2-12640004-P00003. All of the SAXS and TEM measurements were taken at the Materials Characterization Lab (MCL) in the Materials Research Institute (MRI) at the Penn State University. We are grateful to John Cantolina (Huck Institutes of the Life Sciences) for help with cryo-SEM imaging.

## Author contributions

C.L., M.K., and R.J.H. designed the project. C.L. carried out the synthesis of the polymers, and characterized the materials using TEM and tensile tests. J.A.L. conducted the SAXS experiments, and modeled the SAXS patterns. N.U. conducted the rheology experiments, and M.J.N. prepared all of the polymer solutions for the rheology experiments. Y.X. ran all the UV−Vis experiments for the plasmonic hydrogels. W.S. helped C.L. run experiments. M.A.H. helped with printing of the hydrogel materials. M.A.H., R.H.C., and M.K. added valuable insight into the full understanding of the hydrogel formation. C.L. and R.J.H. co-wrote the manuscript with significant input from M.K.

## Additional information

**Competing interests:** The authors declare no competing interests.

