## [Transparent Peer Review File · Nature Communications]

Reviewers' comments:

Reviewer #1 (Remarks to the Author):

The authors report "a rapid-injection method as a versatile and robust technique for preparing hydrogel materials with outstanding mechanical". Injection of ABA block-copolymers into a non-solvent for A block creates hierarchically organized hydrogels with interesting photonic and mechanical properties. Non-equilibrium preparation conditions are suited for continuous and additive manufacturing. The paper is focused on exploration of experimental parameters that govern quenching ABA self-assembly, targeting the establishment of a quantitative route for controlling morphology and properties in a predictive way. Although the big picture is clear and significant correlations have been observed, the paper lacks fundamental understanding of the system. The demonstrated proofs-of-concept of potential applications are incoherent and mostly qualitative. Focusing on a couple of applications would make this paper stronger.

1. Fig. 2e: What is the physical origin of two distinct populations of particles? And, why are the 0.07 micelles larger than micelles formed at higher concentrations?
2. The c^* discussion in Lines 219-238 is trivial and can be either omitted or moved to SI.
3. Figure 3d: What is origin of the non-linear stress increase (strain-stiffening)? Is this a material or morphology effect?
4. Figure 3e: What is the origin of strain-rate dependence?
5. Figure 3e: Are the observed "relaxation steps" reproducible? Do they always occur at the same strain? Do they depend on hydrogel morphology? The abrupt nature of these steps suggests macroscopic material failure rather than microscopic relaxation. This could be due to macroscopic heterogeneity of hydrogels. Please comment.
6. Figure 4 h,i: More information needed.
7. The idea of "a plasmonic or photonic band gap hydrogel" sounds very conceptual and needs more details on the origin and control of color.
8. Line 398: For coating formation, an object should be submerged into water followed by rapid quenching of film morphology. Quickness of the process impedes formation of a uniform layer of a predetermined thickness. Please comment.
9. In Line 403-404: Removal the coating in a single piece suggests weak adhesion. Please comment.

Reviewer #2 (Remarks to the Author):

The authors describe a broadly applicable processing methods by which ABA triblock copolymers produce micellar, microgel or hydrogel materials. They refer to their polymer processing methods as solvent-non-solvent rapid injection, which is akin to precipitation or coagulation of a polymer solution. In this case, the structures are more diverse because the solubility of the ABA triblock in hydrophobic-hydrophilic-hydrophobic in the initial solvent (THF) and final solvent (water). The study is broad in that they have demonstrated these structures in four triblock copolymers. The detailed comments below are meant to further improve and clarify the presentation of their results.

1) A critical aspect of this study is the solvent quality for the A- and B-blocks of the triblock copolymer. The authors have elected to use qualitative descriptions including "good solvent" and "B-selective" solvent. While the hierarchical structures are non-equilibrium, it would be valuable and insightful to have the solvent quality described quantitatively. Two readily available options are the homopolymer-solvent interactions parameters or solubility parameters. This additional information will provide valuable guidance to others attempting to extend these results to other systems.

2) The distinction between microgel and hydrogel needs to be clarified. Structurally microgels and

hydrogels are very similar on the ~ 1 micron length scale with intermicellar bridging of the PEO mid-blocks. The morphological distinction between microgels and hydrogels seems to be the number of intermicellar PEO bridges which increases with polymer concentration, which probably varies as a continuum. At lower polymer concentrations, there are enough bridges to make a weak gel, while at higher polymer concentrations the gels are more robust. This has nothing to do with the length scale as the terms microgel and hydrogel suggest. As Figure 2f suggest, the fraction of intermicellar PEO bridges relative to intramicellar PEO loops depends on both the concentration in THF (c/c^*) as well as the molecular weights of the blocks.

3) Is there a correlation between the water content in the gel and the designation of microgel versus hydrogel?

3) Figure 2f is a bit misleading. By plotting data from all four polymers in the same figure, the authors obscure the fact that some of the polymers have only one or two data points at concentrations below c^* . This jeopardizes the claim that the specific viscosity has a different dependence on c above and below c^* .

4) The authors might be overstating the practical value of their results. Industry tries to avoid solvent processing methods and concerns about residual THF might prohibit the use of these structures in medical applications.

5) For equation 1, the other use the pervaded volume. Please confirm that the R_g measure by light scattering is in THF.

6) The structures are reported to be recyclable, but a better claim would be reversible.

7) The authors might consider simplifying their triblock copolymer notation to eliminate the decimals: IOI (17.3-52.6-17.3) could become IOI (17-53-17). The detailed molecular weights still reported in a table.

Reviewer #3 (Remarks to the Author):

This is an excellent paper, articulating novel work that combines a new route to self-assembled polymer structures with methodologies that lend themselves to new processing techniques. Together with the supporting information and videos, this is a fascinating and compelling piece of work. Generally speaking the characterization work is good, although I don't see the point to insisting that there is a 1.3 exponent to the concentration dependence in Figure 2 and in related Figures in the SI. There may be theory that indicates that but the data do not. I think the paper can be published nearly as is.

Reviewer Comments

Reviewer 1

General Comment: *The authors report “a rapid-injection method as a versatile and robust technique for preparing hydrogel materials with outstanding mechanical”. Injection of ABA block-copolymers into a non-solvent for A block creates hierarchically organized hydrogels with interesting photonic and mechanical properties. Non-equilibrium preparation conditions are suited for continuous and additive manufacturing. The paper is focused on exploration of experimental parameters that govern quenching ABA self-assembly, targeting the establishment of a quantitative route for controlling morphology and properties in a predictive way. Although the big picture is clear and significant correlations have been observed, the paper lacks fundamental understanding of the system. The demonstrated proofs-of-concept of potential applications are incoherent and mostly qualitative. Focusing on a couple of applications would make this paper stronger.*

Response: Although we appreciate the supporting comments from the reviewer, we disagree that the manuscript lacks fundamental understanding. The main point of the work was to explore an easy, robust, and quantitative procedure for creating nanostructured materials from micelles to hydrogels, which has not been previously reported. Furthermore, we have gone into great detail to characterize the system, which presents a full understanding of the structures and key features of the materials. Demonstrated with hydrogel printing, fiber forming, and materials coating with photonic and plasmonic properties, it is highlighted that the new method could enable broad implementation into many areas such as tissue engineering, drug delivery, soft robotics, and optical coatings. Therefore, the proofs-of-concept may seem incoherent, but allows different research communities and the broad readership of *Nature Communications* to use our method for different applications. In terms of specific material properties, systematic studies will be needed for a thorough understanding of all the details, which is not the main focus of this work. According to the comments and questions provided by the reviewer, we have added a number of new experiments and analyses to the paper. However, several of these investigations are ongoing and will likely be described in subsequent papers dedicated to different aspects of the material.

Comment 1: Fig. 2e: What is the physical origin of two distinct populations of particles? And, why are the 0.07 micelles larger than micelles formed at higher concentrations?

Response 1: We are currently investigating the physical origin of the two distinct populations of single micelles and micellar aggregates, but we hypothesize that the triblock copolymer propensity to loop vs bridge under the rapid injection process described in the manuscript is a balance between polymer chain stretching and chain bridging probability distribution. At low polymer concentrations, the rapid injection process leads to the formation of discrete micelles. When the polymer concentration approaches c^* , discrete micelles as well as microgels are formed. We posit that the factor dominating this trend is the probability for a single polymer chain being caught by only one micelle (looping), or by two neighboring micelles (bridging). The probability of a chain to bridge between two different micelles is low when the micelle population is low for concentrations well below c^* . As the polymer concentration increases, the micelle population increases, and the likelihood for a chain to bridge two micelles increases. The role of chain stretching will play a larger role with increasing concentration. The balance between polymer chain stretching and chain bridging probability distribution are the focus of a future publication that will include theory and experiment.

The reason DLS experiments indicate that the micelles formed at low concentration are larger than the micelles formed at high concentrations is due to a modelling artifact of the intensity autocorrelation function. For the low concentration micelle sample, the size is from fitting of a single exponential decay function, while for high concentrations, the size is from fitting a double exponential decay function. For the latter case, it is more difficult to get accurate size value of the discrete micelles that only account for a small portion of the scattering intensity.

To further confirm this conclusion, micelle core diameter for two different micelle samples were determined by analyzing TEM images (**Figure R1**). The TEM analysis indicates that the size of the single micelles is similar for two different samples: single micelles (0.02 wt%, $c/c^* = 0.025$) and micelles in a micelle/microgel mixture (1.0 wt%, $c/c^* = 1.25$). The diameter of the micelles in the 0.025 SOS(8-65-8) sample is 13.5 ± 1.8 nm, while the diameter of the discrete micelles in 1.25 SOS(8-65-8) sample (c/c^*) (“micelle/microgel mixture”) is 15.8 ± 2.7 nm. We would like to note that the TEM images

probe the micelle core diameter in the dry state, while the DLS measurements probe the hydrodynamic diameter that includes the micelle core and corona.

Figure R1. Size analysis of discrete micelles from TEM characterization of SOS(8-65-8) sample. (a, c) 0.02 wt% SOS(8-65-8), $c/c^* = 0.025$ (b, d) 1.0 wt% SOS(8-65-8), $c/c^* = 1.25$.

Comment 2: The c^* discussion in Lines 219-238 is trivial and can be either omitted or moved to SI.

Response 2: The c^* discussion in lines 219-238 is in some way trivial for researchers familiar with polymers. The point of the discussion is to enable researchers from fields outside of polymers to easily create various nanostructures using the method described in the manuscript. Furthermore, *Nature Communications* is an interdisciplinary journal with a broad scope. We therefore have decided to keep the discussion in the main text.

***Comment 3:** Figure 3d: What is origin of the non-linear stress increase (strain-stiffening)? Is this a material or morphology effect?*

Response 3: Strain-stiffening has been observed in many physically associated polymer network systems.¹⁻⁶ While the detailed mechanisms of nonlinear mechanical behavior varies from one system to another, two main theories have been proposed. One is the increased number of elastic active chains during stretching⁶⁻⁹ and the other one is the failure of Gaussian polymer chain assumption at large strain values.^{2, 6, 10-13} Either way, strain-stiffening is considered to be a result of polymer chain network deformation at large strains, and thus to be a property related to the material itself. For the gels created using rapid injection, a more systematic study is needed to determine the exact mechanism governing the non-linear mechanical behaviors, and the role the morphology plays.

***Comment 4:** Figure 3e: What is the origin of strain-rate dependence?*

Response 4: The strain-rate dependence of any polymer material arises from the fact that polymer chains are able to adapt to deformations, if the strain-rate is slower than the characteristic time scale for a polymer chain. For example, polymer materials will exhibit brittle fracture if the strain-rate is faster than the ability for a polymer chain to respond to the deformation. Here, at reduced strain-rates, the polymer chains are able to adapt to the deformation, and as a result, increase the strain at break.

***Comment 5:** Figure 3e: Are the observed “relaxation steps” reproducible? Do they always occur at the same strain? Do they depend on hydrogel morphology? The abrupt nature of these steps suggests macroscopic material failure rather than microscopic relaxation. This could be due to macroscopic heterogeneity of hydrogels. Please comment.*

Response 5: To confirm the reproducibility of the stress-strain behavior of the hydrogel material observed at low strain rate (1/min), additional experiments were performed under the same experimental conditions (hydrogel fiber diameter and polymer concentration). As seen in **Figure R2**, the “relaxation steps” are indeed reproducible. Furthermore, the “relaxation steps” do seem to occur at similar strains, although there are some deviations in the exact strain values. We are currently investigating the origin of

the “relaxation steps” and determining the effect of nanostructure and cellular network structure on the mechanical properties.

We agree with the reviewer that the fiber samples may have heterogeneities, and the “relaxation steps” could be a result of macroscopic material deformation. Thus, simultaneous video characterization was performed on the hydrogel fibers during low-strain-rate tensile test in order to identify the cause of the “relaxation steps”. However, no significant macroscopic deformation (tear, break, or delamination) related to the relaxation steps was observed from the measurements with our current experimental set-up (**Figure R3**). Therefore more thorough and in situ experiments using other in depth characterization techniques (such as digital-image-correlation (DIC) analysis¹⁴⁻¹⁵ and in situ tensile test coupled with X-ray/neutron/electron microscopy¹⁶⁻¹⁸) are needed to identify the exact cause of the “relaxation steps”. **Figures R2** and **R3** have been added to the supporting information. We have updated the sentence on page 12, line 302. Specifically, we state,

“Unexpectedly, the stress-strain curve also shows multiple relaxation steps at low strain rates (**Figure 3e**). Although replicate experiments show these relaxation steps are reproducible and occur at similar strain values (**Figure S9**), the exact cause of these steps is not currently known and further in depth studies are needed to fully address the mechanism.^{15, 17,}”

Figure R2. Stress-strain curves of hydrogel fibers created using 15 wt% SOS(8-65-8) THF solution measured at a strain rate of 1/min.

Figure R3. Images of hydrogel fiber samples from using a 15 wt% SOS(8-65-8) THF solution pulled (1/min) at different elongation ratio values.

Comment 6: Figure 4 h,i: More information needed.

Response 6: Details on the coating conditions were added in the description of **Figure 4h** and **4i**. Specifically, “(h, i) Hydrogel coatings on a faceted glass sphere. The glass sphere is able to be coated with (h) a plasmonic hydrogel layer, or (i) a photonic band gap hydrogel layer. The coatings were accomplished by submerging a cleaned faceted glass sphere into (h) a THF solution of 10 wt% SOS(8-65-8) with AuNPs (10 wt% with respect to polymer), and (i) a THF solution of 10 wt% SOS(69-156-69).”

Comment 7: The idea of “a plasmonic or photonic band gap hydrogel” sounds very conceptual and needs more details on the origin and control of color.

Response 7: To better define the concept of “plasmonic”, we added a few sentences to the main text.

“The plasmonic properties of the gold nanoparticles give rise to the red color seen in the hydrogel materials, which is a result of the collective oscillation of electrons in the nanoparticle.¹⁹ By co-assembling gold nanoparticles and polymers using solution processing techniques, it is possible to create materials exhibiting physical properties that are dictated by the nanoparticle, which is one of the attractive features of colloidal nanoparticle/polymer materials.^{20-22,,}”

Furthermore, we added a sentence to the main text that defines photonic band gap.

“The photonic band gap properties of the hydrogels are a result of the material to reflect or transmit specific wavelengths of light depending on the morphology and the differences in the refractive indices of the different domains.^{23-25,,}”

In the supporting information, we also added a UV-Vis reflection spectrum (**Figure R4**) of the SOS(69-156-69) hydrogel sample to better characterize the photonic band gap properties. As shown in **Figure R4**, the hydrogel can reflect up to 60% of light at wavelength of 350 nm (UVA region). The reflectance also suggests that the SOS(69-156-69) hydrogel reflects violet (380-450 nm) and blue (450-495 nm) light at the visible spectrum. We are currently investigating the origin of the color and how to control the reflected wavelength.

Figure R4. UV-Vis reflection spectrum of the hydrogel exhibiting photonic band gap properties made from quick injection of a THF solution of 10 wt% SOS(69-156-69). The hydrogel can reflect up to 60% of the light at wavelength of 350 nm (UVA region). The reflectance also suggests that at the visible spectrum, the hydrogel reflects both violet (380-450 nm) and blue (450-495 nm) light.

Comment 8: Line 398: For coating formation, an object should be submerged into water followed by rapid quenching of film morphology. Quickness of the process impedes formation of a uniform layer of a predetermined thickness. Please comment.

Response 8: We agree that our current rapid injection method for making hydrogel coatings will lead to coatings with non-uniform surfaces. The surface roughness is likely due to the fast exchange between water and THF. Further research is needed to study different conditions and the key parameters (such as temperature, humidity, and coating speed) during the process to prepare uniform coatings. We would like to highlight that the focus of the manuscript is not on uniform coatings, but how the rapid injection processing method is versatile and can be used under many different conditions.

Comment 9: In Line 403-404: Removal the coating in a single piece suggests weak adhesion. Please comment.

Response 9: We agree with the reviewer that the adhesion between the hydrogel and the substrate is rather weak due to the fact that their interaction is physical attachment. It has been established in previously published works that in order to get strong adhesion between hydrogel and substrate, chemical anchoring to the substrate is required, especially when the substrate is non-porous (glass in this case).²⁶ More studies are needed to look into the enhancement of the coating adhesion force of the gels to different materials for specific applications.

Reviewer 2

General Comment: The authors describe a broadly applicable processing methods by which ABA triblock copolymers produce micellar, microgel or hydrogel materials. They refer to their polymer processing methods as solvent-non-solvent rapid injection, which is akin to precipitation or coagulation

of a polymer solution. In this case, the structures are more diverse because the solubility of the ABA triblock in hydrophobic-hydrophilic-hydrophobic in the initial solvent (THF) and final solvent (water). The study is broad in that they have demonstrated these structures in four triblock copolymers. The detailed comments below are meant to further improve and clarify the presentation of their results.

Response: We thank the reviewer for their recognition of the impact of our work and its breadth.

Comment 1: A critical aspect of this study is the solvent quality for the A- and B-blocks of the triblock copolymer. The authors have elected to use qualitative descriptions including "good solvent" and "B-selective" solvent. While the hierarchical structures are non-equilibrium, it would be valuable and insightful to have the solvent quality described quantitatively. Two readily available options are the homopolymer-solvent interactions parameters or solubility parameters. This additional information will provide valuable guidance to others attempting to extend these results to other systems.

Response 1: We added a table to the supporting information containing polymer-solvent interaction parameter values at 25 °C from calculations using Equation (R1) and previously published work. It has been established that when the interaction parameter χ is smaller than 0.5, the interaction between the solvent and polymer is favorable (good solvent).²⁷⁻²⁸ Whereas, when the interaction parameter is greater than 1, the interaction between the solvent and polymer is unfavorable (poor solvent).²⁹ As indicated from the table, THF is a good solvent for S, I, B and O, while water is only a good solvent for O. Water is a poor solvent for S, I and B. The analysis and table were added to supporting information.

Table R1. Polymer-solvent interaction parameter (25°C).

Polymer	χ_{THF}	χ_{water}
S	0.02 ^a	6.26 ^a
I	0.12 ^a	7.10 ^a
B	0.18 ^a	6.93 ^a
O	0.02 ^a	0.45 ^b

^aValues calculated using solubility parameter of the solvents and homopolymers using Equation R1,

$$\chi = \frac{V_s}{RT} (\delta_p - \delta_s)^2 \quad (R1)$$

Where χ is Flory interaction parameter, V_s is molar volume of the solvent, δ_p and δ_s are the solubility parameters of polymer and solvent respectively. ^bPreviously reported value.³⁰

***Comment 2:** The distinction between microgel and hydrogel needs to be clarified. Structurally microgels and hydrogels are very similar on the ~ 1 micron length scale with intermicellar bridging of the PEO mid-blocks. The morphological distinction between microgels and hydrogels seems to be the number of intermicellar PEO bridges which increases with polymer concentration, which probably varies as a continuum. At lower polymer concentrations, there are enough bridges to make a weak gel, while at higher polymer concentrations the gels are more robust. This has nothing to do with the length scale as the terms microgel and hydrogel suggest. As Figure 2f suggest, the fraction of intermicellar PEO bridges relative to intramicellar PEO loops depends on both the concentration in THF (c/c^*) as well as the molecular weights of the blocks.*

Response 2: This is an excellent point. We agree with the reviewer that at lower concentrations, the micron-sized micellar aggregates (microgels) will have an increased number of looping (which is hard to accurately measure), as compared to gels formed at higher concentrations. As concentration increases, more bridging chains between different micelles forms, which will lead to a more robust gel. The most ideal way of quantifying the system is to directly compare the amount of looping and bridging, which is an ongoing effort in our lab. As a result, less bridging leads to microscopic gels, which is an easier parameter to observe and characterize.

To make this point clearer, we added a sentence to the main text. Specifically,

“Morphologically, the microgels formed in the work presented here resemble the hydrogels at the nanometer scale. The difference between the two structures is the ratio between looping and bridging chains. The micron-sized micellar aggregates, or microgels, will have an increased number of looping chains, as compared to hydrogels formed at higher concentrations. As concentration is increased, more bridging chains between different micelles forms, which leads to more robust gels with higher water contents (**Table S5**). Here, the microgel term is used to classify the transition between isolated micelles and fully-developed hydrogels.”

Comment 3: *Is there a correlation between the water content in the gel and the designation of microgel versus hydrogel?*

Response 3: Yes, there is a correlation between the water content in the microgel and hydrogel. As shown in the table below, the water content of the microgel samples from rapid injection was experimentally measured and compared with hydrogel samples. It was found the microgels have lower water content compared with hydrogels. Although additional studies need to be conducted to determine why the water content for the hydrogels is greater than the microgels, we suspect that the hydrogels contain more water due to the cellular network structure. The table was added into the supporting information (**Table S5**) and referenced in the main text.

Table R2. Water fraction of the microgels and hydrogels prepared using rapid injection.

Polymer	water fraction of hydrogel (wt%)	water fraction of microgel (wt%)
SOS(8-65-8)	94.8	81.9
SOS(69-156-69)	93.1	83.4
BOB(31-70-31)	92.8	77.5
IOI(17-53-17)	93.7	74.6

Comment 3: *Figure 2f is a bit misleading. By plotting data from all four polymers in the same figure, the authors obscure the fact that some of the polymers have only one or two data points at concentrations below c^* . This jeopardizes the claim that the specific viscosity has a different dependence on c above and below c^* .*

Response 3: We understand how this plot can be considered misleading. In response, we have updated the plot in **Figure 2f**. First off, we have added three lines indicating the expected power laws for the dilute, semi-dilute, and entangled regions without forcing a fit with data. The data points from the four polymers scatter around the three power laws, which is standard in universal plots. We also added additional data points to ensure there are at least three data points in each solution regime.

Figure R5. Universal specific viscosity versus normalized polymer concentration (c/c^*) plot for the SOS, IOI, and BOB polymers used in the study. The black lines indicate the expected power laws for dilute, semi-dilute, and entangled concentration regimes.

With the additional data points, we have updated the location of c^* and revised **Table 1** in the manuscript.

Table 1. Triblock copolymer molecular weights, dispersity values, O volume fraction, and experimentally determined and calculated overlap concentration.

Sample	$M_{n,total}$ (kg/mol)	c^*_{cal} (wt%)	c^*_{exp} (wt%)	$c_{e,exp}$ (wt%)
SOS(8-65-8)	68.8	1.0	0.8	5.0
SOS(69-156-69)	222.9	0.6	0.5	3.3
IOI(17-53-17)	54.5	1.4	1.0	6.0
BOB(31-70-31)	120.3	0.7	0.45	5.5

Comment 4: *The authors might be overstating the practical value of their results. Industry tries to avoid solvent processing methods and concerns about residual THF might prohibit the use of these structures in medical applications.*

Response 4: This is a good point. We do realize that industry is trying to avoid organic solvents during processing, but in the work shown here, the ratio of organic solvent used is low (5% relative to water for micelle and microgels preparation and even lower for hydrogels). Besides, the residual organic solvent can be removed from the samples by dialysis or multiple washes in situations where the usage of organic solvent is a concern. Additional modifications to the method are needed if the hydrogels are to be used for in vivo biological applications.

Comment 5: *For equation 1, the other use the pervaded volume. Please confirm that the R_g measure by light scattering is in THF.*

Response 5: Yes, the R_g values for the polymers are measured from polymer solutions dissolved in THF. This detail was highlighted in the footnote of **Table 1**,

“ R_g values were determined using a Wyatt Multi-Angle Light Scattering detector that was attached to the SEC, which was run with THF as the mobile phase.”

Comment 6: *The structures are reported to be recyclable, but a better claim would be reversible.*

Response 6: We decided to use the term recyclable because the gels formed in water are easily re-dissolved and reprecipitated to afford the original triblock copolymers. Furthermore, the triblock copolymers can be reused to fabricate new gels. Yes, the gels contain reversible physical crosslinks, but reversible in the context of polymer materials is typically designated for dynamic/responsive systems. The responsiveness of the materials hasn't been explored in the current work, therefore it's not clear yet to what extent the gel structures are reversible. Thus, recyclable was used instead of reversible in the manuscript.

Comment 7: *The authors might consider simplifying their triblock copolymer notation to eliminate the decimals: IOI (17.3-52.6-17.3) could become IOI (17-53-17). The detailed molecular weights still reported in a table.*

Response 7: We agree that notation of the polymer can be simplified, thank you for the suggestion. This has now been revised throughout the manuscript and supporting information.

Reviewer 3

General Comment: This is an excellent paper, articulating novel work that combines a new route to self-assembled polymer structures with methodologies that lend themselves to new processing techniques. Together with the supporting information and videos, this is a fascinating and compelling piece of work. Generally speaking the characterization work is good, although I don't see the point to insisting that there is a 1.3 exponent to the concentration dependence in Figure 2 and in related Figures in the SI. There may be theory that indicates that but the data do not. I think the paper can be published nearly as is.

Response: We thank the reviewer for their positive comments. We have updated **Figure 2** in the main text (see Response 3 to Reviewer 2's comment) and the figures in the SI to indicate the theoretical power laws for the three concentration regimes. The data points from the four polymers scatter around the three power laws, which is expected in universal plots. Although the reviewer states that the semi-dilute regime does not seem to follow a scaling law of 1.3, we do see that there is a general trend.

Figure R5. Universal specific viscosity versus normalized polymer concentration (c/c^*) plot for the SOS, IOI, and BOB polymers used in the study.

Reference

- (1) Lemaitre, J., *Handbook of Materials Behavior Models, Three-Volume Set: Nonlinear Models and Properties*. Elsevier: 2001.
- (2) Erk, K. A.; Henderson, K. J.; Shull, K. R., Strain stiffening in synthetic and biopolymer networks. *Biomacromolecules* **2010**, *11*, 1358-1363.
- (3) Erk, K. A.; Shull, K. R., Rate-dependent stiffening and strain localization in physically associating solutions. *Macromolecules* **2011**, *44*, 932-939.
- (4) Hashemnejad, S. M.; Kundu, S., Nonlinear elasticity and cavitation of a triblock copolymer gel. *Soft Matter* **2015**, *11*, 4315-4325.
- (5) Vatankhah-Varnosfaderani, M.; Daniel, W. F.; Everhart, M. H.; Pandya, A. A.; Liang, H.; Matyjaszewski, K.; Dobrynin, A. V.; Sheiko, S. S., Mimicking biological stress-strain behaviour with synthetic elastomers. *Nature* **2017**, *549*, 497.
- (6) Xu, D.; Craig, S. L., Strain hardening and strain softening of reversibly cross-linked supramolecular polymer networks. *Macromolecules* **2011**, *44*, 7478-7488.
- (7) Tirtaatmadja, V.; Tam, K.; Jenkins, R., Superposition of oscillations on steady shear flow as a technique for investigating the structure of associative polymers. *Macromolecules* **1997**, *30*, 1426-1433.
- (8) Tung, S.-H.; Raghavan, S. R., Strain-stiffening response in transient networks formed by reverse wormlike micelles. *Langmuir* **2008**, *24*, 8405-8408.
- (9) Bossard, F.; Sfika, V.; Tsitsilianis, C., Rheological properties of physical gel formed by triblock polyampholyte in salt-free aqueous solutions. *Macromolecules* **2004**, *37*, 3899-3904.
- (10) Pellens, L.; Gamez Corrales, R.; Mewis, J., General nonlinear rheological behavior of associative polymers. *Journal of rheology* **2004**, *48*, 379-393.
- (11) Mewis, J.; Kaffashi, B.; Vermant, J.; Butera, R., Determining relaxation modes in flowing associative polymers using superposition flows. *Macromolecules* **2001**, *34*, 1376-1383.
- (12) Serero, Y.; Jacobsen, V.; Berret, J.-F.; May, R., Evidence of nonlinear chain stretching in the rheology of transient networks. *Macromolecules* **2000**, *33*, 1841-1847.
- (13) Orakdogan, N.; Erman, B.; Okay, O., Evidence of strain hardening in DNA gels. *Macromolecules* **2010**, *43*, 1530-1538.
- (14) Chu, T.; Ranson, W.; Sutton, M. A., Applications of digital-image-correlation techniques to experimental mechanics. *Experimental mechanics* **1985**, *25*, 232-244.
- (15) Moerman, K. M.; Holt, C. A.; Evans, S. L.; Simms, C. K., Digital image correlation and finite element modelling as a method to determine mechanical properties of human soft tissue in vivo. *Journal of biomechanics* **2009**, *42*, 1150-1153.
- (16) Yuan, Z.; Dai, Q.; Cheng, X.; Chen, K.; Pan, L.; Wang, A., In situ SEM tensile test of high-nitrogen austenitic stainless steels. *Materials characterization* **2006**, *56*, 79-83.
- (17) Zhu, Y.; Moldovan, N.; Espinosa, H. D., A microelectromechanical load sensor for in situ electron and x-ray microscopy tensile testing of nanostructures. *Appl. Phys. Lett.* **2005**, *86*, 013506.
- (18) Tomota, Y.; Tokuda, H.; Adachi, Y.; Wakita, M.; Minakawa, N.; Moriai, A.; Morii, Y., Tensile behavior of TRIP-aided multi-phase steels studied by in situ neutron diffraction. *Acta Materialia* **2004**, *52*, 5737-5745.
- (19) Link, S.; El-Sayed, M. A., Size and temperature dependence of the plasmon absorption of colloidal gold nanoparticles. *The Journal of Physical Chemistry B* **1999**, *103*, 4212-4217.
- (20) Hickey, R. J.; Haynes, A. S.; Kikkawa, J. M.; Park, S.-J., Controlling the self-assembly structure of magnetic nanoparticles and amphiphilic block-copolymers: from micelles to vesicles. *J. Am. Chem. Soc.* **2011**, *133*, 1517-1525.
- (21) Hickey, R. J.; Koski, J.; Meng, X.; Riggleman, R. A.; Zhang, P.; Park, S.-J., Size-controlled self-assembly of superparamagnetic polymersomes. *ACS Nano* **2014**, *8*, 495-502.
- (22) Hickey, R. J.; Meng, X.; Zhang, P.; Park, S.-J., Low-dimensional nanoparticle clustering in polymer micelles and their transverse relaxivity rates. *ACS Nano* **2013**, *7*, 5824-5833.

- (23) Joannopoulos, J. D.; Johnson, S. G.; Winn, J. N.; Meade, R. D., Photonic crystals: molding the flow of light, 2008. *Princeton Univ Pr.*
- (24) Yablonovitch, E., Inhibited spontaneous emission in solid-state physics and electronics. *Phys. Rev. Lett.* **1987**, *58*, 2059.
- (25) John, S., Strong localization of photons in certain disordered dielectric superlattices. *Phys. Rev. Lett.* **1987**, *58*, 2486.
- (26) Yuk, H.; Zhang, T.; Lin, S.; Parada, G. A.; Zhao, X., Tough bonding of hydrogels to diverse non-porous surfaces. *Nat. Mater.* **2016**, *15*, 190.
- (27) Hiemenz, P. C.; Lodge, T. P., *Polymer chemistry*. CRC press: 2007.
- (28) Mark, J. E., *Physical properties of polymers handbook*. Springer: 2007; Vol. 1076.
- (29) D. Sudduth, R., A review of the similarities and differences between five different polymer-solvent interaction coefficients. *Pigment & Resin Technology* **2013**, *42*, 394-405.
- (30) Dormidontova, E. E., Role of competitive PEO- water and water- water hydrogen bonding in aqueous solution PEO behavior. *Macromolecules* **2002**, *35*, 987-1001.

REVIEWERS' COMMENTS:

Reviewer #1 (Remarks to the Author):

I agree with authors reply. However, I would like to see changes in the manuscript in response to comments 1, 3, and 4.

Reviewer #3 (Remarks to the Author):

OK to publish as is

Point-by-point response to reviewer comments

Reviewer 1

Comment: I agree with authors reply. However, I would like to see changes in the manuscript in response to comments 1, 3, and 4.

Response: We have made several changes to the manuscript and supporting information to add in our responses to comments 1, 3 and 4.

Comment 1: Fig. 2e: What is the physical origin of two distinct populations of particles? And, why are the 0.07 micelles larger than micelles formed at higher concentrations?

Response 1: To explain the micelle size difference, we have added the following explanation into the manuscript,

“The single micelle size variations seen in DLS for 0.05 wt% and 0.4 wt% samples are attributed to a fitting artifact of the intensity autocorrelation function when there are more than one population sizes, which is confirmed using TEM characterizations (Supplementary Figure 5).”

We’ve also added the following figure into the supporting information.

Figure R1. Size analysis of discrete micelles from TEM characterization of SOS(8-65-8) sample. (a, c) 0.02 wt% SOS(8-65-8), $c/c^* = 0.025$ (b, d) 1.0 wt% SOS(8-65-8), $c/c^* = 1.25$. The diameter of the

micelles in the 0.025 SOS(8-65-8) sample is 13.5 ± 1.8 nm, which is similar to the diameter (15.8 ± 2.7 nm) of the discrete micelles in 1.25 SOS(8-65-8) sample (c/c^*) (“micelle/microgel mixture”).

Comment 3: Figure 3d: What is origin of the non-linear stress increase (strain-stiffening)? Is this a material or morphology effect?

Response 3: To better express our understandings regarding strain-stiffening phenomenon, we have added the following sentences,

“Strain-stiffening has been observed in many physically associated polymer network systems. Two main theories regarding their mechanisms have been proposed: 1) failure of the Gaussian polymer chain assumption at large strain values and 2) Increased number of elastically effective chains between crosslinks during stretching. A more systematic study is needed to determine the exact mechanism governing the non-linear mechanical behaviors of the gel network created by rapid injection, and the role the hierarchical morphology plays.”

Comment 4: Figure 3e: What is the origin of strain-rate dependence?

Response 4: To address this question, we have stated,

“At low strain rate, the polymer chains in the hydrogel networks have more time to adapt to deformation, and as a result exhibit increased elongation at break values.”

Reviewer 2

General Comment: OK to publish as is.

Response: We thank the reviewer for their recognition of the quality of our work and revisions.